# Chemical Defense against Herbivory in the Brown Marine Macroalga *Padina gymnospora* Could Be Attributed to a New Hydrocarbon Compound

**DOI:** 10.3390/plants12051073

**Published:** 2023-02-28

**Authors:** Renato Crespo Pereira, Wladimir Costa Paradas, Rodrigo Tomazetto de Carvalho, Davyson de Lima Moreira, Alphonse Kelecom, Raoni Moreira Ferreira Passos, Georgia Correa Atella, Leonardo Tavares Salgado

**Affiliations:** 1Departamento de Biologia Marinha, Instituto de Biologia, Universidade Federal Fluminense, Niterói 24220-900, Brazil; 2Instituto de Pesquisas Jardim Botânico do Rio de Janeiro, Rio de Janeiro 22460-030, Brazil; 3Departamento de Biologia Geral, Instituto de Biologia, Universidade Federal Fluminense, Niterói 24220-900, Brazil; 4Instituto de Bioquímica Médica, Universidade Federal do Rio de Janeiro, Rio de Janeiro 21941-901, Brazil

**Keywords:** calcification, phlorotannin, lipid, fatty acid

## Abstract

Brown marine macroalga *Padina gymnospora* (Phaeophyceae, Ochrophyta) produces both secondary metabolites (phlorotannins) and precipitate calcium carbonate (CaCO_3_—aragonite) on its surface as potential defensive strategies against herbivory. Here, we have evaluated the effect of natural concentrations of organic extracts (dichloromethane—DI; ethyl acetate—EA and methanol—ME, and three isolated fractions) and mineralized tissues of *P. gymnospora* as chemical and physical resistance, respectively, against the sea urchin *Lytechinus variegatus* through experimental laboratory feeding bioassays. Fatty acids (FA), glycolipids (GLY), phlorotannins (PH) and hydrocarbons (HC) were also characterized and/or quantified in extracts and fractions from *P. gymnospora* using nuclear magnetic resonance (NMR) and gas chromatography (GC) coupled to mass spectrometry (CG/MS) or GC coupled to flame ionization detector (FID) and chemical analysis. Our results showed that chemicals from the EA extract of *P. gymnospora* were significantly important in reducing consumption by *L. variegatus*, but the CaCO_3_ did not act as a physical protection against consumption by this sea urchin. An enriched fraction containing 76% of the new hydrocarbon 5*Z*,8*Z*,11*Z*,14*Z*-heneicosatetraene exhibited a significant defensive property, while other chemicals found in minor amounts, such as GLY, PH, saturated and monounsaturated FAs and CaCO_3_ did not interfere with the susceptibility of *P. gymnospora* to *L. variegatus* consumption. We suggest that the unsaturation of the 5*Z*,8*Z*,11*Z*,14*Z*-heneicosatetraene from *P. gymnospora* is probably an important structural characteristic responsible for the defensive property verified against the sea urchin.

## 1. Introduction

For a little over 20 years, studies in marine chemical ecology have reaffirmed the performance of various molecular types of secondary metabolites of marine macroalgae as chemical defense against herbivores. Through experimental approaches, several terpenoid types of green [1,2], brown [3,4] and red [5,6] macroalgae have been evidenced as a chemical defense against different types of herbivores, such as sea gastropods, mollusks, urchins, fishes and others; while phlorotannins played a role in protecting only brown macroalgae, also against varied types of herbivores [7]. However, on a smaller scale or with rare examples, other molecular types of brown macroalgae have also been evidenced as a defense against herbivores, such as hydrocarbons [8], terpenoid bromoquinone [9], and fatty acids [10]. Conversely, other chemicals from marine brown macroalgae stimulate consumption by herbivores, such as glycolipids [11].

*Padina* (Phaeophyceae, Ochrophyta) species are brown macroalgae also known to produces a diverse array of secondary metabolites, such as terpenes [12,13], hydrocarbons [14], sterols, highly unsaturated and unsaturated fatty acids [15], sulfur-containing compounds [16], phlorotannins [17] and glycolipids [18]. Chemicals from *Padina gymnospora* are known to exhibit several biological activities, such as antibacterial and prominent natural wound-care products [19], and the alpha bisabolol compound inhibited cholinesterase [20]. In addition, a few direct or indirect evidences revealed that chemicals from *Padina* also exhibited ecological roles, such as the fatty acids from *P. tetrastromatica* which had antifouling properties [21], the alpha bisabolol from *P. gymnospora* which also had antifouling properties [22] and extracts from *Padina* species as defensive or feeding-deterrents against snails and abalones gastropods [15]. Other examples include the extracts of *P. tenuis* which inhibited the consumption by the fish *Zebrasoma flavescens* [10] and non-polyphenolic or non-polar secondary metabolites present in the extract of *Padina* sp. inhibited fish communities in the field [23].

Calcification is also an important defensive structural component in the relationship between various marine macroalgae and herbivores, since calcified species are the most conspicuous in coral reef habitats characterized by intense herbivory pressure [24,25]. Several lines of evidence from experimental approaches have correlated the low preference of herbivores for heavily calcified macroalgae, such as fishes [26,27], sea-hare [28], and sea-urchin [29]. Although not increasing the toughness of the macroalgae, calcium carbonate (either calcite or aragonite) have inhibited consumption by the sea hare *Dolabella auricularia* [28]. They can also exert indirect effects by decreasing the nutritional value of the food [30] or causing negative physiological effects on some herbivores [31].

*Padina* species also exhibit calcification (CaCO_3_) as an extracellular matrix, occurring as aragonite needles in alternating concentric bands, mainly at the surface of adaxial parts of the thalli [32,33]. Considering the minimal force required for limpets species to remove its tissue, *Padina* species were classified as lightly calcified and its calcification seems not to be used as structural defense [28,34]. The normal force required to remove tissue by the limpet species *Collisella tranquebarica*, *Tectura elegans* and *Tectura albicosta* decreases as the calcification increases, but secondary metabolites or other attributes probably influence the algae consumption [34]. Chemical and microscopic analyses have evidenced that mineralized and non-mineralized regions of *P. gymnospora* possess different contents of phlorotannins and physical properties, such as deformation, adhesion, topography and nano-rugosity [35], which presupposes a possible distinct relationship of these parts of the thallus of *Padina* with consumers.

The present study specifically addressed the following questions: (1) Do secondary metabolites of *P. gymnospora* act as a defense against consumption by the sea urchin *Lytechinus variegatus*?; (2) which substance of *P. gymnospora* would be a chemical defense against *L. variegatus*?; (3) what is the susceptibility of mineralized (CaCO_3_-containing) and non-mineralized tissues of *P. gymnospora* to *L. variegatus*?

## 2. Results

### 2.1. Feeding Assays

Dichloromethane (DI) and ethylacetate (EA) extracts of *P. gymnospora* were significantly less consumed by *L. variegatus* compared to their respective controls, and both significantly less consumed than the methanol (ME) extract (Figure 1, *p* < 0.05, ANOVA). However, no significantly differential consumption was found between consumed ME extract food and its respective control (Figure 1, *p* > 0.05, ANOVA). The solvent had no effect on these results, since the solvent- (CTRL ME) and non-solvent (CTRL)-containing foods were equally consumed by *L. variegatus* (Figure 1). Among the fractions from the EA extract, only FA3 was significantly less consumed than its respective control (Figure 2, *p* < 0.05, ANOVA), while FA1 and FA2 were not consumed differently from their controls (Figure 2, *p* > 0.05, ANOVA).

Mineralized (MIN) and demineralized (DEM) tissues of *P. gymnospora* were less consumed than their respective controls (green macroalga *U. fasciata*) (Figure 3, *p* < 0.05, ANOVA). However, both tissue-types of *P. gymnospora* were not consumed in a significantly different way by *L. variegatus* (Figure 3, *p* > 0.05, ANOVA).

### 2.2. Chemical Analyses

The chemical structure of the *P. gymnospora* major compound isolated from FA3 was suggested by NMR (^1^H, APT experiments, HSQC and HMBC), GC/MS, GC/FID, TLC and RI analysis and comparison of our data with that from the literature [36,37,38,39,40]. It was identified as the new compound 5*Z*,8*Z*,11*Z*,14*Z*-heneicosatetraene (C21:4) (Figure 4), a hydrocarbon (HC) similar to 1*Z*,6*Z*,9*Z*,12*Z*,15*Z*-heneicosapentaene (C21:5) previously isolated from the brown macroalga *Fucus vesiculosus* [40]. This new compound is an isomer of 1*E*,3*Z*,6*Z*,9*Z*-heneicosatetraene (C21:4) that was already identified in the moth *Utetheisa ornatrix* [38,39].

The ^1^H NMR spectra of FA3 displayed resonances of a ill-resolved triplet attributed to terminal methyl group(s), complex hydrogens signals of an aliphatic chain at δ 1.24, 1.40 and 2.02; triplets of bis-allylic methylene groups at δ 2.82, 2.83 and 2.85; and olefinic hydrogens at δ 5.35 (*dt*, *J* = 10.6 and 6.1 Hz, Hd) (Appendix A). Both, the chemical shifts and coupling constant of these olefinic hydrogens indicated the presence of disubstituted double bonds in the *Z* configuration. The correlation in the HSQC spectrum between the signal at ~δ 128 and δ 5.35 corroborated this assumption.

A similar ^1^H NMR profile was published for the synthetic unsaturated hydrocarbon 1*Z*,3*Z*,6*Z*,9*Z*-heneicosatetraene (C21:4) [38,39]. GC/MS analysis for FA3 major compound showed some fragments, including *m*/*z* = 164 and *m*/*z* = 175; (Figure 5A) that were not recorded in the mass spectrum of the already described 1*Z*,3*Z*,6*Z*,9*Z*-heneicosatetraene: *m*/*z* = 39, 41, 43, 53, 54, 55, 57, 66, 68, 71, 77, 78, 79, 80, 81, 83, 91, 92, 93, 94, 95, 105, 106, 133, 234, 288 [38]. FA3 major compound (5*Z*,8*Z*,11*Z*,14*Z*-heneicosatetraene) showed fragments at *m*/*z* = 40, 41, 55, 67, 79, 91, 105, 119, 133, 150, 164, 175, 190, 203, 215, 229, 243 (Figure 5A). The fragments for the homologue 5*Z*,8*Z*,11*Z*,14*Z*-eicosatetraenoic acid or araquidonic acid (C20:4) *m*/*z* = 27, 41, 55, 77, 79, 91, 105, 119, 133, 150, 166, 177, 193, 206, 304 are shown in Figure 5B. The similarity between both mass data (Figure 5A,B) strongly supports that 5*Z*,8*Z*,11*Z*,14*Z*-heneicosatetraene and arachidonic acid possess an identical molecular fragment of at least ca. 164 mass units as common structural characteristic. In addition, successive losses of 14 mass units related to CH_2_ (*m*/*z* = 91, 105, 119 and 133 and *m*/*z* = 215, 229 and 243) indicated the presence of the aliphatic chain.

The calculated retention indices (RI_C_) of the FA3 major compound were obtained using three distinct methods: FMS of FA without esterification = 2029; MS of FA esterification = 2041; and from FID of FA without esterification = 2027 (Appendix A). The RI_C_ of the major compound in FA3 agrees with literature data for an unsaturated HC with C21:4 (RI_L_, 2021) [37]. In addition, when the FA3 and FA standards chromatograms were compared, the peak of FA3 major HC (C21:4) did not match with any known FA peak (Figure 6A,B). As mentioned before, the NIST library suggested 5*Z*,8*Z*,11*Z*,14*Z*-eicosatetraenoic acid or arachidonic acid as the main compound in FA3 (Figure 5B). In this way, NIST information guided the double bond positions at C-5,8,11,14 mainly because of the ions at *m*/*z* = 164 and 175 for 5*Z*,8*Z*,11*Z*,14*Z*-heneicosatetraene (Figure 5A) that are similar to *m*/*z* = 166 and 177 for 5*Z*,8*Z*,11*Z*,14*Z*-eicosatetraenoic acid or arachidonic acid (Figure 5B). This search in the library showed a great structural similarity between both HC, but as previously indicated, the major HC is a C21:4 (considering mass fragments and RI_C_), such as this new compound eluted in GC/MS of the esterification products before the oleic (C18:1, *n*-9) and the stearic acids (C18:0) (Figure 6B). The base peaks in the MS of 5*Z*,8*Z*,11*Z*,14*Z*-heneicosatetraene were registered at *m*/*z* = 79/91 (Figure 5A), and no molecular peak could be observed, probably because of its high instability at 70 eV (Figure 5A). The molecular ion at *m*/*z* = 288 [M.+] of the synthetic compound 1*Z*,3*Z*,6*Z*,9*Z*–heneicosatetraene (C21:4) has been previously determined [38].

Considering all datasets presented and the fact that FA3 was extracted from a TLC spot with an Rf of 0.86, characteristic of HC [40], we have suggested that FA3 is composed of a mixture of FA and HC, but the major compound of this fraction was identified as the new natural unsaturated hydrocarbon 5*Z*,8*Z*,11*Z*,14*Z*-heneicosatetraene (Figure 4, Table 1).

The GC/MS analyses of the *P. gymnospora* extracts allowed the identification of 23 different FAs (Appendix A), with palmitic acid (C16:0) > palmitoleic acid (C16:1) > oleic acid (C18:1, *n*-3) as the three major FAs. The major polyunsaturated FAs (PUFAs) were identified as linoleic acid (C18:2, *n*-6), arachidonic acid (C20:4, *n*-6) and eicosapentaenoic acid (C20:5, *n*-3). *Padina gymnospora* FA1 and FA2 showed similar FAs profiles when compared to the crude extracts DI, EA and ME: palmitic acid > oleic acid > stearic acid. Instead, the *P. gymnospora* FA3 had a distinct composition: eight compounds were identified in a decreasing order: 5*Z*,8*Z*,11*Z*,14*Z*-heneicosatetraene > palmitic acid > stearic acid > oleic acid > palmitoleic acid > myristic acid 14:0 > pentadecanoic acid 15:0 (Table 1).

The PERMANOVA analyses determined significant differences regarding the FAs derivative compositions among samples (DI, EA, ME, FA1, FA2 and FA3, *p* < 0.001, PSEUDO-F = 123.16 and df = 5 (Appendix A). The pair-wise test showed that these differences are basically found in the FA3 and in the other samples (DI, EA, ME, FA1, FA2, *p* > 0.05 (Appendix A).

Spectrophotometric analysis has revealed a higher PH amounts (0.23% ± 0.03 DW—dry weight) in ME extracts from *P. gymnospora*, compared with a quantity found in DI (0.01% ± 0.001 DW, *p* = 0.0024) and EA (0.03% ± 0.01 DW, *p* = 0.0028) extracts. No significant differences in phlorotannin contents were registered between DI and EA extracts (*p* = 0.3598). The PH amounts in *P. gymnospora* natural populations ranged between 0.11% ± 0.01 to 0.38% ± 0.02 DW.

Glycolipids TLC densitometry analyses showed that *P. gymnospora* ME extracts presented a higher content of SFL (sulfoquinovosyldiacylglycerols—SQDG, digalactosyldiacylglycerols—DGDG) and CMH (ceramide monohexoside) GLY than DI and EA extracts (*p* = 0.0001). Monogalactosyldiacylglycerol (MGDG) did not vary among ME, DI and EA extracts (*p* = 0.5695). Finally, both MGDG and DGDG were found in significantly higher content in DI and EA extracts than SQDG and GLY (*p* = 0.0001).

## 3. Discussion

In the present study, we have investigated whether the marine brown macroalga *Padina gymnospora* exhibits chemical defense against consumption by the sea-urchin *Lytechinus variegatus*. Using an experimental laboratory approach, we have evidenced that the dichloromethane (DI) and ethyl acetate (EA) extracts show a defensive property against *L. variegatus*. To our knowledge, this is the first demonstration that *P. gymnospora* has a specific compound as a chemical defense against herbivory. However, it reaffirms t that the *Padina* species can be chemically defended against consumers, since extracts of *Padina crassa*, *P. australis* and *P. japonica* inhibited consumption by the abalones *Haliotis discus hannai*, *H. discus discus* and *H. gigantea*, and the snails *Chlorostoma lischkei*, *Omphalius rusticus* and *O. pfeifferi carpenteri* [15]. Additional evidence includes the defensive action of a *P. tenuis* extract against the fish *Zebrasoma flavescens* [10] and non-polyphenolic or non-polar secondary metabolites of *Padina* sp. That can inhibit fish communities in the field [23].

Among the three fractions obtained from the defensive EA extract, only one of them inhibited the consumption by *L. variegatus* and chemical analyses of it has revealed the new hydrocarbon 5*Z*,8*Z*,*11Z*,14*Z*-heneicosatetraene (C21:4) as the major compound, with traces of fatty acids (FA) in the relative proportions 18:0 > 18:1 *n*-9 > 16:1 > 14:0 > 15:0 > 19:0. This new hydrocarbon (HC) 5*Z*,8*Z*,11*Z*,14*Z*-heneicosatetraene has a structure and configuration which is identical to its homologue 5*Z*,8*Z*,11*Z*,14*Z*-eicosatetraenoic acid or arachidonic acid (C20:4 *n*-6). Some previous studies have also shown that macroalgal HCs derived from arachidonic acid and eicosapentaenoic acid (EPA, C20:5 *n*-3) pathways act as a chemical defense against herbivores [8,42,43,44,45]. For example, C_11_ sulfur metabolites from *Dictyopteris* spp. deterred grazing by the amphipod *Ampithoe longimana*, but exhibited low defensive effect on the consumption by the sea urchin *Arbacia punctulata* [42]. Also C_11_ HCs were more deterrent to *A. longimana* grazing than to *A. punctulata* grazing [8].

Hydrocarbons (HCs) are compounds which are also commonly produced by other species of marine brown macroalgae, such as *Pilayella littoralis* (*cis*-3,6,9,12,15-heneicosahexaene, 21:6) and *Ascophyllum nodosum* (*cis*-3,11-heptadecadiene, 17:2) [37], as well as *Scytosiphon lomentarea* (*cis*-4,7,10,13-nonadecatetraene, 20:4; and *cis*-3,6,9,12,15-nonadecapentaene, 20:5), *Fucus distichus* and *F. vesiculosus* (*cis*-1,6,9,12,15,18-heneicosahexaene, 20:6) and *Laminaria saccarina* (*cis*-1,6,9,12,15-heneicosapenatene, 20:5) [36] and supposedly could be a chemical defense against herbivory.

Feeding assays have also revealed that the methanolic extract (ME) of *P. gymonospora* was the most consumed by *L. variegatus* and contained low amounts of phlorotannins (PH). Our data showed that a low PH content in the *P. gymnospora* ME extract (0.23% ± 0.03 DW) used in feeding assays and from samples of the natural populations (0.11% ± 0.01 to 0.38% ± 0.02 DW) agree with the literature for tropical brown macroalgae (<0.5%) [35,46,47]. That said, these amounts are very low to inhibit herbivory [47]. For example, the PH extracted from tropical *S. furcatum* were a deterrent against the amphipods *Parhyale hawaiensis* only at concentrations of 2 and 5% in artificial food, but the natural amount (0.5%) did not deter feeding by this amphipod [47]. In this way, our data agree with some previous studies using temperate brown macroalgae that showed that phlorotannins are not used as chemical defense against herbivory [48,49].

The *P. gymnospora* ME extracts also contained sulfoquinovosyl diacylglycerols (SQDG), glycosphingolipids (GLY) and digalactosyldiacylglycerols (DGDG), but in higher amounts than in DI and EA extracts. Some studies have shown that isolated GLYs are macroalgal phagostimulants [11,18,50], while another studies have shown that isolated GLYs have inhibited herbivory [48]. *DGDG* enriched with unsaturated FAs from *Padina arborecens* have stimulated consumption by the marine gastropod *Haliotis discus* [18], as well as DGDG and 1,2-diacylglyceryl-4′-*O*-(*N*,*N*,*N*-trimethyl)-homoserine (DGTH) isolated from ME extract from *Ulva pertusa* have stimulated consumption by *H. discus* [50]. The present data lead us to infer that *P. gymnospora* GLY are probably phagostimulant components of the ME extract, but further studies with these isolated compounds are necessary to confirm this assumption.

The major FAs identified in *P. gymnospora* were palmitic acid (16:0), palmitoleic acid (16:1) and oleic acid (18:1, *n*-9), where the major PUFA were linoleic acid (C18:2, *n*-6), arachidonic acid (C20:4, *n*-6) and eicosapentaenoic acid (C20:5, *n*-3). These data followed the same trend reported earlier for Ochrophyta (brown algal species) [16,51,52,53]. Tabarsa et al. (2012) [54] found a similar FA profile for *Padina pavonica*, where the major FAs and PUFAs were, respectively, 16:0, 16:1/18:1 and 20:4 (*n*-6)/20:5 (*n*-3). A similar FA pattern was also evidenced in *P. pavonica* [16], in which the major FAs (16:0, 18:1 and 14:0) and PUFAs (20:4, *n*-6; 20:5, *n*-3) were identified and the major FAs as 16:0 and 18:1 identified in *P. vickersiae* [51].

The fraction FA3 isolated from *P. gymnospora*, which inhibited *L. variegatus* consumption, presented among its components, beyond the major compound 5*Z*,8*Z*,11*Z*,14*Z*-heneicosatetraene, FAs such as myristic acid (C14:0), pentadecanoic acid (C15:0), palmitoleic acid (C16:1), oleic acid (C18:1, *n*-9) and stearic acid (C18:0). Highest defensive property was found in a fraction from the extract of the diatom *Diatoma tenuis* containing the unsaturated FAs *cis-*5,8,11,14,17-eicosapentaenoic acid against grazing by *Thamnocephalus platyurus* [55]. In the same study, other active isolated fractions active against *T. platyurus* contained hexadecadienoic acid, α-linolenic acid (18:3), palmitoleic acid (16:1) and myristic acid (14:0) [55]. Since the *P. gymnospora* FA3 contains other traces of FAs, such as 14:0, 15.0, 16:1, 18:1 (*n*-9) and 18:0, we infer that these compounds could also be contributing to the observed defensive activity against *L. variegatus*.

Studies have shown the importance of unsaturation for the defensive property against herbivores and pathogens. The unsaturated FA from *Turbinaria ornata* (Ochrophyta) 20-hydroxy-4,8,13,17-tetramethyl-4,8,12,16-eicosatetraenoic acid inhibited consumption by two herbivores *Omphalius pfeifferi* and *Turbo marumoratus* [56]. The eicosapentaenoic acid (20:5, *n*-3) from the diatom *Phaeodactylum tricornutum* and other FAs such as palmitoleic acid (C16:1) and 6*Z*,9*Z*,12*Z*-hexadecatrienoic acid (C16:3 *n*-4), inhibited the growth of the Gram-negative marine bacteria fish pathogen *Listonella anguillarum* [5]. The 20:5 (*n*-3), also from *P. tricornutum*, inhibited the growth of a multi-resistant strain of *Staphylococcus aureus* (MRSA) at micromolar concentrations [57]. Long-chain unsaturated acids, such as palmitoleic, oleic, linolenic and arachidonic acids inhibited bacterial enoyl-acyl carrier-protein reductase (FabI), an essential component of bacterial FA synthesis [58].

In the present study, saturated FAs were detected as major chemicals in non-defensive extracts or fractions evaluated against *L. variegatus*, while fractions enriched with the unsaturated 5*Z*,8*Z*,11*Z*,14*Z*-heneicosatetraene inhibited the consumption by this sea urchin. Previous studies have shown that unsaturated FAs exhibited more effective defensive property than saturated FAs against bacteria [59,60,61] and against herbivores [48,55,56]. These data about defensive activity of unsaturated molecules led us to suggest that the unsaturation portion of the major compound 5*Z*,8*Z*,*11Z*,14*Z*-heneicosatetraene found in *P. gymnospora* would be important for the defensive action verified here against the *L. variegatus*.

Regarding the ecological role of the calcification (CaCO_3_) as physical protection of *P. gymnospora*, both MIN and DEM tissues of this brown macroalga were not differentially consumed by *L. variegatus*. The calcified species *Padina tenuis* was also readily eaten by the sea hare *Dolabella auricularia*, probably due to its soft thallus being easily bitten by this mollusc [28]. Their low susceptibility to ingestion by this sea urchin was evidenced by the fact that *P. gymnospora* individuals were less consumed by *L. varigatus* than the corresponding control *U. fasciata*. Similar results were also previously reported about *Padina durvillei* which was less consumed by the sea urchin *Echinometra vanbrunti* than *Ulva rigida* in field trials [24]. In fact, the use of calcium carbonate as protection against herbivory is far from a consensus, not only for *Padina* species. For example, for two highly calcified species *Corallina vancouveriensis* and *Corallina officinalis* var. *chilensis*, the reductions in calcium carbonate content did not cause a significant increase in urchin grazing [62]. Alternatively, the mechanism by which CaCO_3_ inhibits herbivory may simply be due to the decreased nutritional value of the macroalgae or chemicals that stimulate the consumption [28]. This second possibility may be true for *P. gymonospora* studied here, since it exhibits FA and GLY that could stimulate the consumption by *L. variegatus* and overrides the effect of CaCO_3_. The obtained results lead us to infer that in *P. gymnospora* chemicals probably provided by the new compound HC 5*Z*,8*Z*,11*Z*,14*Z*-heneicosa-tetraene is more important than the physical one (CaCO_3_ mineralization) to defend this macroalga against *L. variegatus*.

## 4. Materials and Methods

### 4.1. Samples

*Padina gymnospora* specimens were collected from the intertidal zone at Rasa beach (Rio de Janeiro State; Brazil; 22°43′58″ 41°57′25″ W). After collection, living macroalgal samples were stored in filtered seawater inside a dark isothermal chamber and transported to the laboratory. Thereafter, *P. gymnospora* individuals were maintained inside an 8 L aquarium with seawater enriched with Provasoli medium [63], under controlled conditions as already described elsewhere [35].

### 4.2. Feeding Bioassays

Feeding assays were carried out using an echinoid species, the generalist consumer *Lytechinus variegatus*, which feeds on marine algae [64], and usually avoids food items that possess structural aspects and/or chemical defenses [65].

Two kinds of *L. variegatus* feeding bioassays were carried out: (1) evaluation of defensive effect of natural concentrations of *P. gymnospora* extracts (dichloromethane, DI; ethyl acetate, EA; methanol, ME) and isolated fractions (FA_1_, FA_2_ and FA_3_) against this sea-urchin in artificial foods; (2) susceptibility of mineralized (MIN) and demineralized (DEM) tissues of *P. gymnospora* to this sea-urchin.

For the assays 1, artificial foods were prepared according to the usual method [66]. The artificial Control foods were prepared by adding 0.72 g of agar to 20.0 mL of distilled water, heating in a microwave oven until boiling point. This mixture was then added to 16.0 mL of distilled water containing 2.0 g of freeze-dried *Ulva* sp. (Chlorophyta), a highly preferred food item [42]. Control food (without extracts or isolated fractions), but with solvent (CTRL ME) and without solvents (CTRL), were also prepared in order to make sure that any eventual solvent residue was not an artifact interfering in the bioassay result. Treatment foods were similarly prepared, but the crude extract or fraction was first dissolved in CH_2_Cl_2_ and added to the 2.0 g of freeze-dried *Ulva* sp. and then the solvent was removed by rotary evaporation. This procedure was necessary to obtain a uniform coating of the metabolite on the algal particles prior to addition to agar [43] before adding and following the food preparation.

Before the bioassays, individuals of *L. variegatus* were maintained in a recirculating laboratory aquarium at constant temperature (20 °C), salinity (35 PSU) and aeration. After an acclimation of 24 h, the bioassays were carried out. Treatments and controls were hardened into a nylon screen and cut into small pieces (10 × 10 squares), which were then simultaneously offered to the sea urchin *L. variegatus* (*n* = 10). The defensive property was estimated by comparing the number of consumed squares between treatment artificial foods (DI, EA, ME, FA_1_, FA_2_, FA_3_ and solvent—ME) and controls foods with (WS) and without solvents (WTS).

For the second bioassay-type (2), mineralized (MIN) pieces of *P. gymnospora* tissues were treated with HNO_3_ 5% (3 × 30 min) for obtaining demineralized tissue (DEM) according to the usual method [35]. Both tissues (MIN and DEM) were washed in filtered seawater, inserted in Petri dishes and inspected with a stereomicroscope (Olympus SZX7, Tokyo, Japan) [35]. Afterwards, the water excess in MIN and DEM tissues of *P. gymnospora* and respective control (*Ulva* sp.) thallus was removed using a filter paper to obtain the wet weight of each tissue. Treatment (MIN or DEM) and respective control were simultaneously offered to the sea urchin *L. variegatus* (*n* = 10). After the end of the assay, the water excess from treatments and controls was removed again to obtain their wet weight. The mean difference between treatment and control was expressed as percentage (%) of consumption. Specimens of *L. variegatus* were maintained in the laboratory aquarium under the conditions as described in the first assay.

### 4.3. Chemical Analyses

In order to extract metabolites with distinct polarities, three crude extracts were obtained from *P. gymnospora*: dichloromethane (DI), ethyl acetate (EA), and methanol (ME) solvents Merck (Readington Township, NW, USA). The yields were DI 0.8%, EA 0.9% and ME 1.2% (*w*/*w*). *Padina gymnospora* DI, EA and ME were submitted to identification of fatty acid FA and glycolipids—GLY, and quantification of phenolic substances (PH = phlorotannins). Fatty acids were identified using GC/MS according to the usual procedures [67], phlorotannins (PH) were quantified using the Folin–Ciocalteau (FC) method [68] and the GLY using TLC in silica gel 60 aluminum sheets (Merck & Co. Inc., Readington Township, NW, USA) [69].

Three fractions were obtained from the EA extract, that were solubilized in methanol (Merck (Readington Township, NW, USA), using preparative one-dimensional TLC silica gel 60 F_254_S (Merck, Readington Township, NW, USA) for neutral lipids with the following mobile phase: hexane, diethyl ether and acetic acid (Merck, Readington Township, NW, USA) at 90:7.5:1 (*v*/*v*/*v*) [44]. These fractions were analyzed for the lipid content (FAs) using GC/MS, but only FA derivatives (FAs) and hydrocarbons (HC) were found. For this purpose, the isolated fractions from *P. gymnospora* EA were named FA1, FA2 and FA3, because they showed mainly fatty acid derivatives in its constitution with TLC and GC/MS analysis. These fractions presented the following TLC reference factors (Rfs): FA1 (0.20), FA2 (0.52) and FA3 (0.86). The yields were 0.6%, 0.5% and 0.1%, respectively.

For FAs analyses, 1 mg/mL of the *P. gymnospora* extracts (*n* = 3) and fractions (*n* = 3) were submitted to esterification: samples were dissolved in toluene (C_7_H_8_, Merck, Readington Township, NW, USA), and treated overnight, at 50 °C, with a 1% sulphuric acid (H_2_SO_4_) solution in methanol (Merck, Readington Township, NW, USA). After that, a 5% aqueous sodium chloride (NaCl) solution was added to the reaction medium and the esters were extracted with hexane (Merck, Readington Township, NW, USA), using a Pasteur pipette to collect separated phases. The hexane layer was washed with 2% potassium bicarbonate (KHCO_3_) in distilled water, and the mixture was evaporated under a nitrogen-saturated atmosphere. Thereafter, the sample was re-suspended in 50 μL of hexane, and a 1 μL aliquot was analysed using GC/MS. The analyses were performed in a Shimadzu QP2010 Plus GC instrument coupled to a Mass Spectrometry Detector (Shimadzu Corporation, Kyoto, KR, Japan), equipped with a Hewlett-Packard Ultra 2 polysiloxane capillary column (Hewlett-Packard Company, Palo Alto, CA, USA) (25 m × 0.20 mm i.d. × film thickness 0.33 μm). The injector temperature was maintained at 250 °C, and 1 μL aliquots were injected in the split mode ratio of 1:1. The column oven temperature was programmed to increase from 40 °C to 160 °C at 30 °C/min; and from 160 °C to 233 °C at 1 °C/min; and from 233 °C to 300 °C at 30 °C/min. After that, temperature was maintained at 300 °C for 10 min. Helium was used as a carrier gas at a constant flow rate of 1 mL/ min. Electron impact spectra were recorded in positive mode at 70 eV with a scan time of 1 s. Mass fragments were detected in full scan mode from 40 to 600 (*m*/*z*). The FA compounds in *P. gymnospora* extracts and fractions were identified by comparing their mass spectra with the mass spectra of FAME 37-methylated FA mix standards (Supelco, Sigma-Aldrich Company, Saint Louis, MO, USA) and the standard series of *n*-alkanes (C_7_–C_30_, Sigma-Aldrich Company, Saint Louis, MO, USA) obtained on the same equipment in identical conditions. Two injections were performed with (*n* = 3) and without esters extraction. The Retention Indices (RIs) of the FA3 were determined relative to the retention times of a series of *n*-alkanes (C_7_–C_30_) with linear interpolation. GC/MS software version 2.53 (Shimadzu Corporation, Kyoto, Japan) was used for data processing.

The FA3 fraction was also analyzed in a Shimadzu GC 2010 (Shimadzu Corporation, Kyoto, Japan) coupled to a Flame Ionization Detector (FID), equipped with a DB5 (Agilent J & W, Santa Clara, CA, USA) fused silica capillary column (30 m × 0.25 mm i.d. × film thickness 0.25 μm). The oven temperature was programmed to 50 °C to 240 °C at 3 °C min/min, then hold at 240 °C for 20 min. Injector and detector temperatures were set and maintained at 220 °C and 290 °C, respectively. An analyzed sample was dissolved in CHCl_3_ (Merck, Readington Township, NW, USA), and 1 μL aliquots were injected in the split mode with a ratio of 1:40 using H_2_ as the carrier gas (1.44 mL/min). The relative amounts of the components were calculated based on GC peak areas without correction factors.

FA_3_ was also analyzed with 1D and 2D NMR techniques (NMR; ^1^H and ^13^C/400 MHz and 500 MHz, CD_3_OD.

Both *P. gymnospora* extracts obtained with solvents of different polarities (DI, EA and ME; *n* = 5) and extracts from natural populations (*n* = 5) (which were submitted to acetone:water, 7:3, extraction) were analyzed using the FC method [45]. The quantification of PSs was performed by adding 1N FC reagent (Sigma-Aldrich, Saint Louis, MO, USA) to a 400 μL aliquot of diluted extract (100 μg/mL). The quantification was performed in a spectrophotometer Libra S-80 (Biochrom, Cambridge, UK) at 750 nm, using a calibration curve obtained with a phloroglucinol standard (Sigma-Aldrich, Saint Louis, MO, USA) at 10 μg/mL, 20 μg/mL, 30 μg/mL and 40 μg/mL (ABS = 0.1021 × Conc − 0.1197; *r*^2^ = 0.99). PSs based on phloroglucinol content are represented as % of dry weight.

The *P. gymnospora* DI, AE and ME extracts were submitted to GLY analyses using TLC according to the usual method [46]. The solvent system used was chloroform: methanol: ammonium hydroxide 2 M (40:10:1). Sulfoglycolipids (sulfatides, SFL) and ceramide monohexosides (CMH) standards were from Sigma-Aldrich Company (Saint Louis, MO, USA). Based on the Rfs the following lipids were identified: sulfoquinovosyl diacylglycerols (SQDG), glycosphingolipids (GLY), monogalactosyl diacylglycerol (MGDG) and digalactosyl diacylglycerol (DGDG). The lipids were quantified by analysing digital images of silica plates, with the Image Master TotalLab software (Nonlinear Dynamics Limited, Newcastle, UK).

### 4.4. Statistical Uni- and Multivariate Analyses

Analyses of the results from the *L. variegatus* feeding bioassays with artificial foods containing *P. gymnospora* extracts (DI, EA, ME; *n* = 5), fractions (FA_1_, FA_2_ and FA_3_; *n* = 5) and fronds (MIN/DEM) were performed with a one-way ANOVA (with *post hoc* Tukey). The percentages of FAs derivatives (% of content) were compared among treatments (DI, EA, ME, FA_1_, FA_2_ and FA_3_, *n* = 3) by the analysis of variance (PERMANOVA) with a Euclidean distance matrix and 999 permutations (significant results, *p* < 0.05).

To compare FAs compositions among treatments (DI, EA, ME, FA1, FA2 and FA3, *n* = 3), chromatogram peak areas (% of content) of the compounds obtained using GC/MS of each treatment were compared using Bray–Curtis similarity (Cluster and principal component analyses, PCA). FA data matrix included all the FAs derivatives reported in the Appendix A. The one-way ANOVA (*post hoc* Tukey) was also used to evaluate the differences of PH and GLY levels among *P. gymnospora* extracts (DI, EA and ME; *n* = 5). Significant results were confirmed when *p* < 0.05 (α = 5%). Univariate statistical analyses were performed using STATISTICA software (version 6.0; StatSoft, Inc., Tulsa, OK, USA), and multivariate statistical analyses were conducted using the PRIMER software program (version 6.0; PRIMER-E Ltd., Ivybridge, UK).

## Figures and Tables

**Figure 1 plants-12-01073-f001:**
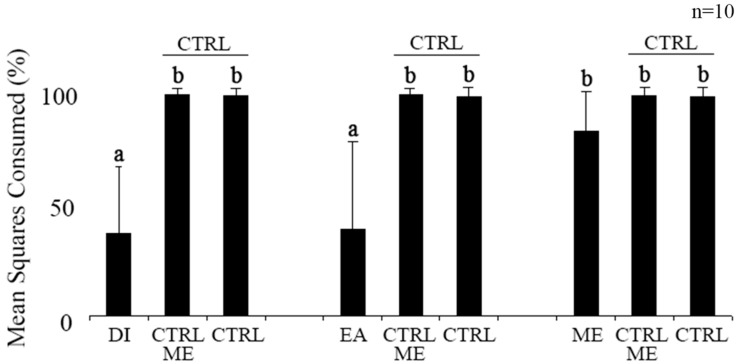
Effect of *P. gymnospora* extracts (dichloromethane DI, ethyl acetate EA and methanol ME) on the consumption by the sea urchin *L. variegatus*. Controls: CRTL, only grounded *U. fasciata* with solvent (CTRL ME) and without solvents (CTRL). Different letters above bars indicate distinct consumption (*p* < 0.05). Values are means and standard deviation of *n* = 10.

**Figure 2 plants-12-01073-f002:**
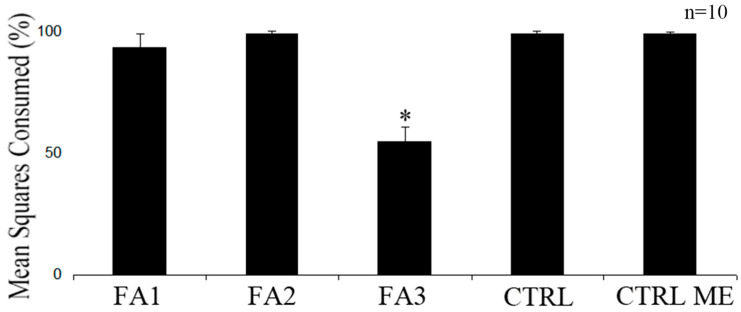
Effect of fatty acid fractions (FA1, FA2 and FA3) from EA extract of *P. gymnospora* on the consumption by *L. variegatus*. Controls: only grounded *U. fasciata* with (CTRL ME) and without methanol (CTRL). Asterisk indicate distinct consumption (*p* < 0.05). Values are means and standard deviation of *n* = 10.

**Figure 3 plants-12-01073-f003:**
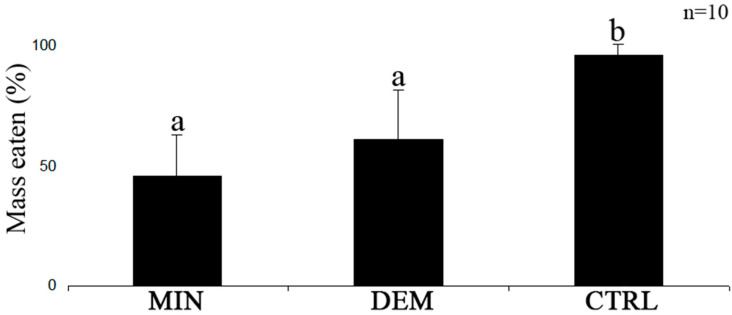
Mean percentage (%) mass eaten of mineralized (MIN) and demineralized (DEM) tissues of *P. gymnospora* and control (CTRL—*U. fasciata*) by *L. variegatus*. Different letters indicate distinct consumption (*p* < 0.05). Values are means and standard deviation of *n* = 10.

**Figure 4 plants-12-01073-f004:**
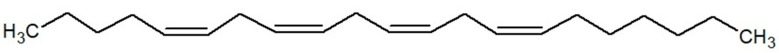
Chemical structure of the new compound 5*Z*,8*Z*,11*Z*,14*Z*-heneicosatetraene isolated from *P. gymnospora*.

**Figure 5 plants-12-01073-f005:**
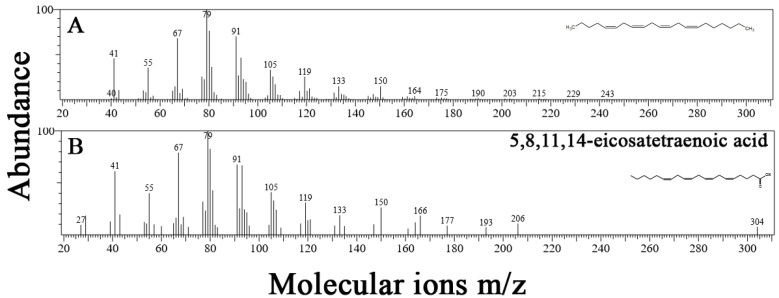
(**A**) Mass (MS) data of *P. gymnospora* FA3 major peak (attributed to 5*Z*,8*Z*,11*Z*,14*Z*-heneicosatetraene). (**B**) The NIST suggestion for FA3 major peak (5Z,8Z,11Z,14Z-eicosatetraenoic acid or arachidonic acid, C20:4).

**Figure 6 plants-12-01073-f006:**
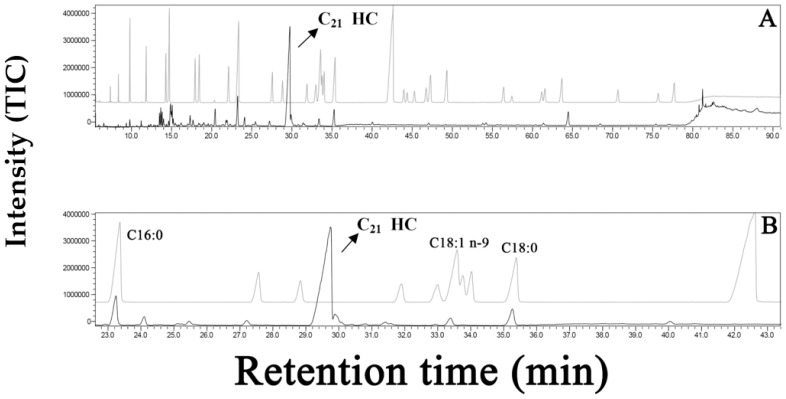
Chromatograms (CG/MS) of the fraction FA3 from *P. gymnospora* in (**A**) Low magnification; (**B**) Higher magnification. The chromatogram of the major compound 5*Z*,8*Z*,11*Z*,14*Z*-heneicosatetraene (C21:4 HC, (**B**)) was superimposed with a FAs standards chromatogram obtained in the same conditions [(**B**) C16:0 = palmitic acid, C18:1 (*n*-9) = oleic acid, and C18:0 = stearic acid]. TIC = total ion chromatogram.

**Table 1 plants-12-01073-t001:** Gas chromatography coupled to Mass Spectrometry (GC/MS) analysis of fatty acids (FAs) and hydrocarbons (HC) from *P. gymnospora* in the fraction FA3. RI = Retention index. The FAs identification was made comparing the mass spectrum of FAME FAs and the calculated Retention Index (RIc in HP, 25 m × 0.20 mm i.d. × 0.33 μm, MS) with those from literature (RI_L_) for FAs (a) [41] and HCs (b) [37]. Results are presented as mean (%) ± standard deviation (*n* = 3). RT = Retention time.

Peak	Compound (FA and HC)	RIc	RI_L_	Molecular Formula	Lipid	FA3	
						RT	%
1	Myristic acid	1723	1713 ^a^	C_14_H_23_O_2_	14:0	14.6	1.14 ± 0.10
2	Pentadecanoic acid	1825	1813 ^a^	C_15_H_30_O_2_	15:0	18.4	0.71 ± 0.05
3	Palmitoleic acid	1902	1888 ^a^	C_16_H_30_O_2_	16:1	21.9	1.79 ± 0.12
4	Palmitic acid	1926	1913 ^a^	C_16_H_32_O_2_	16:0	23.2	9.80 ± 1.10
5	5*Z*,8*Z*,11*Z*,14*Z*-Heneicosatetraene	2041	2021 ^b^	C_21_H_36_	21:4	29.7	76.60 ± 5.10
6	Oleic acid	2099	2081 ^a^	C_18_H_34_O_2_	18:1 (*n*-9)	33.4	3.02 ± 0.15
7	Stearic acid	2127	2113 ^a^	C_18_H_36_O_2_	18:0	35.3	6.87 ± 0.30

## Data Availability

The complete data collected in the research are available in the article and Appendix A.

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
