# Peer review of "Chemical Defense against Herbivory in the Brown Marine Macroalga Padina gymnospora Could Be Attributed to a New Hydrocarbon Compound"

_plants, 2023, doi:10.3390/plants12051073_

Round 1

Reviewer 1 Report

The manuscript analyzes the role of secondary metabolites of the brown algae, Padina gymnospora, against the sea urchin Lytechinus variegatus. For this purpose different algae extracts were applied on the food of the sea urchin and its feeding was analyzed. The authors could show a significant activity for the dichloromethane and the ethyl acetate extracts. For the ethyl acetate extracts the activity could be further narrowed down to a particular subfraction that beside other compounds is enriched in a new Heneicosatetraene compound that was identified with different analytical methods including GC-MS and NMR. Additionally, the authors analyzed the role of mineralization on sea urchin feeding and couldn’t observe any effect on the sea urchin feeding.

Major comments:

While the presence of some bioactive compounds can be concluded from the presented data, the actual function of the Heneicosatetraene is only slightly indicated. The manuscript however overemphasizes the defensive function of the new Heneicosatetraene on various occasions, in particular in the title. The presented data only show that the compound is enriched in an active subfraction in one out of two active extracts. Just being one of the most abundant compounds in an active fraction doesn’t prove its role for the activity. Similarly, other compounds can’t be excluded as active compounds in that way. For this more support for example from a more specific manipulation would be necessary. 

I don’t fully get the point (4) from the last paragraph of the introduction and how this is addressed by the presented data.

 The abstracts mentions that “extracts enriched with GLY, PH, and saturated and monounsaturated FAs stimulated the L. variagtus consumption”. Where can the respective data be found?

 Figures with GC-MS chromatograms and MS fragmentation patterns shouldn’t be labeled with “Abundance” on the y-axis, but rather something like signal intensity since the sensitivity of different analytes might differ and therefore can’t be directly compared regarding their abundance.

Figure 6 What is shown GC-MS or GC-FID data?

In case of MS chromatograms it is not clearly stated if a total ion chromatogram is shown or if the data for a particular fragment were extracted (same applies for the data in the supplementals).

 Was the new Heneicosatetraene also present in other extracts and fractions than FA3. This compounds seems to be missing in Table S1, and Table 1 only shows the data for FA3.

The manuscript should be checked for language issues. In particular the Discussion should be more carefully revised since it contains a lot of language mistakes and confusing wording e.g., the paragraph from line 241-251, line 297 “against antibacteria“, line 301 “Regarding to ecological of the calcification“ , line 304 “probable due to its the soft thallus being easily torn and bitten”, … and so on.

Figures should use a consistent formatting in respect to front size. What do the error bars show? Standard error or deviation? What does 100% correspond too (WS or WTS)? Were DI, EA and ME experiments conducted in parallel or separate. Was the statistic conducted over the whole dataset or for the 3 solvent types and respective controls separately?

line260 Which particular data indicate that P. gymnospora GLY are probably phagostimulants if the literature seems to be inconsistent?

 Please provide a more detailed description of the method for the fractionation to obtain FA1, FA2 and FA3. E.g. in which solvent the fractions were resuspended. 

Minor comments:

line 77 and 210 Change “Padina gymnospora” to “P. gymnospora” as it was introduced before

line100 Word repetition (“of of”)

line 100 – 102, 122, 145, 169 Species names should be italic

line 106-107 “only grounded U. fasciata with and without methanol (CTRL ME)” Specify abbreviation for both treatments

line 149 Word repetition (“from from”)

line 218 “0. pfeifferi carpenter” there seems to be zero instead of an “O”

line 349 “salinity (35)” unit missing

line 496 Correct typo to “PostDoc”

Supplemental data 1 Correct typo “Ressonance”

Table S1 – Positioning in the tables seems to be a bit shifted 

Please also correct the formatting of the References. Exemplary issues are listed below:

line 571 correct „quali- ty”

line 576 remove „740“

line 612 Correct the formatting of the Journal name (italic, abbreviation); Year in wrong position and should be bold

line 631 Species names should be italic

line 632 Year should be bold

Author Response

Reviewer 1

The manuscript analyzes the role of secondary metabolites of the brown algae, Padina gymnospora, against the sea urchin Lytechinus variegatus. For this purpose different algae extracts were applied on the food of the sea urchin and its feeding was analyzed. The authors could show a significant activity for the dichloromethane and the ethyl acetate extracts. For the ethyl acetate extracts the activity could be further narrowed down to a particular subfraction that beside other compounds is enriched in a new Heneicosatetraene compound that was identified with different analytical methods including GC-MS and NMR. Additionally, the authors analyzed the role of mineralization on sea urchin feeding and couldn’t observe any effect on the sea urchin feeding.

Major comments:

While the presence of some bioactive compounds can be concluded from the presented data, the actual function of the Heneicosatetraene is only slightly indicated. The manuscript however overemphasizes the defensive function of the new Heneicosatetraene on various occasions, in particular in the title. The presented data only show that the compound is enriched in an active subfraction in one out of two active extracts. Just being one of the most abundant compounds in an active fraction doesn’t prove its role for the activity. Similarly, other compounds can’t be excluded as active compounds in that way. For this more support for example from a more specific manipulation would be necessary. 

Answer: We agree that the reviewer is right to mention that the manuscript “overemphasizes the defensive function of the new heneicosatetraene on various occasions”. In fact, we have not tested heneicosatetraene pure and in its natural concentration found in P. gymnospora as a defense against the sea-urchin L. variegatus. However, there is some indirect evidence supporting the defensive action of this new metabolite, as mentioned below:

- Most of the marine chemical ecology studies have shown that the major compounds are just those that express ecological roles, such as defense against consumers or competitors; very rare are the cases of defensive action evidenced for minor compounds.

- This book mentions several examples of major macroalgal compounds as a chemical defense:

Amsler, CD. 2008. Algal chemical ecology. Springer-Verlag Berlin Heidelberg.

- A comparative analysis revealed that the substance heneicosatetraene is only present in FA3 (Table 1), but was not found in the fractions FA1 and FA2 (Table S1, supplementary);

- In fractions F2 and F3 another abundant substance was found, palmitic acid, with 39.08% and 25.26%, respectively. However, these fractions did not inhibit consumption by L. variegatus;

- In fraction F1, the new compound represents a high percentage of 78.60% and the second most abundant substance palmitic acid represents only 9.80%, but even more abundant in fractions F2 and F3, this chemical did not inhibited herbivory.

            For these reasons, we have chosen to maintain the presumed defensive action of heneicosatetraene, but we have minimized this proposition at various occasions in the text, including the change in the title of the article.

- I don’t fully get the point (4) from the last paragraph of the introduction and how this is addressed by the presented data.

Answer: This question has been modified.

- The abstracts mentions that “extracts enriched with GLY, PH, and saturated and monounsaturated FAs stimulated the L. variagtus consumption”. Where can the respective data be found?

Answer: This aspect was minimized in the article, since the extracts did not significantly stimulate consumption.

- Figures with GC-MS chromatograms and MS fragmentation patterns shouldn’t be labeled with “Abundance” on the y-axis, but rather something like signal intensity since the sensitivity of different analytes might differ and therefore can’t be directly compared regarding their abundance.

Answer: It is right. We can name MS fragmentation pattern with “abundance”, that means total percentage of a detected ion.

- Figure 6 What is shown GC-MS or GC-FID data?

Answer: Our analyses show GC-MS y-axis as intensity (TIC, that means total ion chromatogram) and for GC-FID intensity (that means detector intensity that is based on the electrical conductivity generated by ions after flame combustion). Considering these, GC-MS and GC-FID intensity scales can cannot be compared. Figures were corrected.

- In case of MS chromatograms it is not clearly stated if a total ion chromatogram is shown or if the data for a particular fragment were extracted (same applies for the data in the supplementals).

Answer: No particular fragment was extracted. The GC-MS shows all fragments from scanning range of masses (m/z) at 40-600. This information was included in the text.

- Was the new Heneicosatetraene also present in other extracts and fractions than FA3. This compounds seems to be missing in Table S1, and Table 1 only shows the data for FA3.

Answer: Heneicosatetraene was only found in fraction F3 (Table 1), but was not detected in fractions 2 and 3 (Table S1, supplementary). We did not choose to leave the data for fraction 1 in Table 1 in the text of the article and those for fractions 2 and 3 as supplementary material, since they are not primary targets of the article.

- The manuscript should be checked for language issues. In particular the Discussion should be more carefully revised since it contains a lot of language mistakes and confusing wording e.g., the paragraph from line 241-251, line 297 “against antibacteria“, line 301 “Regarding to ecological of the calcification“ , line 304 “probable due to its the soft thallus being easily torn and bitten”, … and so on.

Answer: These aspects were carefully reviewed.

- Figures should use a consistent formatting in respect to front size. What do the error bars show? Standard error or deviation? What does 100% correspond too (WS or WTS)? Were DI, EA and ME experiments conducted in parallel or separate. Was the statistic conducted over the whole dataset or for the 3 solvent types and respective controls separately?

Answer: The font size has been standardized. Standard deviation in the legend of the figures was included. We have replaced WS and WTS with CTRL ME and CTRL to standardize the figure legends. The experiments were performed separately (DI, EA and ME and respective controls), but simultaneously. Statistical analysis was done separately; each solvent compared to its respective control.

- Line 260 Which particular data indicate that P. gymnospora GLY are probably phagostimulants if the literature seems to be inconsistent?

Answer: The literature is consistent about the stimulating effect of GLY, but we decided to minimize this approach in the article.

- Please provide a more detailed description of the method for the fractionation to obtain FA1, FA2 and FA3. E.g. in which solvent the fractions were resuspended. 

Answer: We strong believe that the experimental to achieve FA1… FA3 It is very well explained: “Three fractions were obtained from EA extract through preparative one-dimensional TLC silica gel 60 F254S (Merck, Readington Township, NW, USA) for neutral lipids with the following mobile phase: hexane, diethyl ether and acetic acid (Merck, Readington Township, NW, USA) at 90:7.5:1 (v/v/v)44. These fractions were analyzed for the lipid content (FAs) by GC/MS, but only FA derivatives (FAs) and hydrocarbons (HC) were found. For this purpose, the isolated fractions from P. gymnospora EA were named FA1, FA2 and FA3, because they showed mainly fatty acid derivatives in its constitution by TLC and GC/MS analysis.

EA extract = from text “… In order to extract metabolites with distinct polarities, three crude extracts were obtained from P. gymnospora: dichloromethane (DI), ethyl acetate (EA), and methanol (ME) solvents Merck (Readington Township, NW, USA).”

Even so, we insert in the text that the EA fraction was solubilized in methanol.

So… new text = “… Three fractions were obtained from EA extract, that was solubilized in methanol (Merck (Readington Township, NW, USA), by one-dimensional TLC silica gel 60 F254S (Merck, Readington Township, NW, USA) for neutral lipids…”

- Minor comments:

- line 77 and 210 Change “Padina gymnospora” to “P. gymnospora” as it was introduced before

Answer: This aspect has been corrected.

- Line100 Word repetition (“of of”) Answer: This aspect has been corrected.

Answer: This duplicity has been removed.

- Line 100 – 102, 122, 145, 169 Species names should be italic

Answer: This aspect has been corrected.

- Line 106-107 “only grounded U. fasciata with and without methanol (CTRL ME)” Specify abbreviation for both treatments.

Answer: We have made these abbreviations and also corrected them in the text (lines 348-349).

- Line 149 Word repetition (“from from”)

Answer: This duplicity has been removed.

- Line 218 “0. Pfeifferi carpenter” there seems to be zero instead of an “O”

Answer: This aspect has been corrected.

- Line 349 “salinity (35)” unit missing

Answer: We included PSU, since salinity was measured in an electronic equipament.

- Line 496 Correct typo to “PostDoc”

Answer: This aspect has been corrected.

- Supplemental data 1 Correct typo “Ressonance”

Answer: This word has been corrected.

- Table S1 – Positioning in the tables seems to be a bit shifted 

Answer: The table was redone and the positioning was corrected.

- Please also correct the formatting of the References. Exemplary issues are listed below: 

- line 571 correct „quali- ty”

- line 576 remove „740“

- line 612 Correct the formatting of the Journal name (italic, abbreviation); Year in wrong position and should be bold

- line 631 Species names should be italic

- line 632 Year should be bold

Answer: The aspects pointed out were corrected and a thorough review of all references was performed

Reviewer 2 Report

The manuscript titled “A new hydrocarbon compound, rather than other chemicals, is the chemical defense against herbivory in the brown marine macroalga Padina gymnospora” reports findings of some importance.

Overall, the manuscript is well written, however, there are some issues which are mentioned below.

The abstract is somewhat well written, however, it lacks in important data.  

The introduction is specific and focused on. The last paragraph of the “Introduction” described what the authors intend to do but this should be revised to make the objectives clear, robust and concise.

Materials and methods section is well written. However, some grammatical errors were spotted.

Results are quite interesting and analysis is strong; well written and explained.

Discussion confirmed results very well and is a logical explanation thereof. However, some of the statements are non sequitur.

Conclusion needs revision. The authors should give some recommendation on the basis of their findings.

Numerous stylistic errors were also spotted.

References are adequate and need to be crosschecked.

The language is up to the mark; however, some grammatical errors were spotted. In some cases, the authors used present tense to describe the results.

Author Response

Reviewer 2

The manuscript titled “A new hydrocarbon compound, rather than other chemicals, is the chemical defense against herbivory in the brown marine macroalga Padina gymnospora” reports findings of some importance. 

Overall, the manuscript is well written, however, there are some issues which are mentioned below. 

- The abstract is somewhat well written, however, it lacks in important data.  

Answer: Without knowing what the data is, it is difficult for us to make the correction. However, the summary has been carefully reviewed in order to verify the absence of data that has been obtained.

- The introduction is specific and focused on. The last paragraph of the “Introduction” described what the authors intend to do but this should be revised to make the objectives clear, robust and concise.

Answer: The objectives were revised.

- Materials and methods section is well written. However, some grammatical errors were spotted. 

Answer: The language of the article has been carefully revised.

- Results are quite interesting and analysis is strong; well written and explained.

Answer: Although no improprieties were noted by your review, the results were carefully reviewed.

- Discussion confirmed results very well and is a logical explanation thereof. However, some of the statements are non sequitur. 

Answer: The discussion has been carefully reviewed in order to minimize undue assertions or extrapolations.

- Conclusion needs revision. The authors should give some recommendation on the basis of their findings.

Answer: The article has been carefully revised.

- Numerous stylistic errors were also spotted. 

Answer: The article has been carefully revised.

- References are adequate and need to be crosschecked. 

Answer: the references were carefully reviewed for the format required by the journal and were also crosschecked.

- The language is up to the mark; however, some grammatical errors were spotted. In some cases, the authors used present tense to describe the results.

Answer: The results have been revised and all data is in the past tense.

Round 2

Reviewer 1 Report

Minor comments:

Please carefully check the paragraph line 296-302. It sounds a bit like some words might be mixed up there.

Author Response

We have revised and rephrased all the sentences that of this paragraph, in order to make the information clearer or more objective.
